# Exploring health-seeking behavior for non-communicable chronic conditions in northern Bangladesh

**Fatema Binte Rasul** [1,2] *, **Malabika Sarker** [1,2], **Farzana Yasmin** [3], **Manuela De Allegri** [1]

**1** Medical Faculty and University Hospital, Heidelberg Institute of Global Health, Ruprecht-Karls-Universität Heidelberg, Heidelberg, Germany, **2** BRAC JPG School of Public Health, BRAC University, Shahid Tajuddin Ahmed Sharani, Mohakhali, Dhaka, Bangladesh, **3** Faculty of Health, Department of Human Medicine, University Witten/Herdecke, Witten, Germany

* fatema.rasul85@gmail.com

## Abstract

Non-communicable Diseases (NCDs) account for 67% of total deaths in Bangladesh. However, the Bangladeshi health system is inadequately prepared to tackle NCDs. Evidence on NCD-specific health-seeking behavior can help appropriately address the needs of people affected by NCDs in Bangladesh. Our study aims to explore health-seeking behavior for people affected by NCDs in northern Bangladesh. We conducted a qualitative study in Mithapukur, Rangpur, during 2015–2016. We purposely selected respondents and carried out 25 in-depth interviews with individuals affected by non-communicable diseases and 21 healthcare providers. Additionally, we held six focus group discussions in the wider community. We verbatim transcribed all interviews and analyzed the content using thematic analysis, according to the following thematic areas: individual, household, and contextual factors that influence health-seeking behavior for NCDs within the context of the broader socio-economic environment. Study findings indicate that people seek care only when symptoms disrupt their daily lifestyle. Henceforth, people's health beliefs, religious beliefs, and relations with local providers direct their actions, keeping provider accessibility, cost anticipation, and satisfying provider-encounters in mind. Health-seeking is predominantly delayed and fragmented. Semi-qualified providers represent a popular first choice. Gender roles dominate health-seeking behavior as women need their guardian's permission to avail care. Our findings indicate the need to sensitize people about the importance of early health-seeking for NCDs, and continuing life-long NCD treatment. Our findings also highlight the need for people-centered care, making preventive and curative NCD services accessible at grassroots level, along with relevant provider training. Furthermore, special provisions, such as financial support and outreach programs are needed to enable access to NCD care for women and the poor.

**Data Availability Statement:** All qualitative data relevant to this manuscript have been included as quotes in this manuscript. The transcripts are in Bangla, and they are not publicly available due to

them containing information that could compromise research participant's privacy, but their key quotes have been translated to English and made available in this manuscript. The other relevant information such as the tools for IDIs and FGDs have been included as supplementary information.

**Funding:** MDA: Costs for field data collection were covered using bonus research funding assigned directly to Prof. Manuela De Allegri (MDA) by the University Hospital Heidelberg to cover research costs for her team. FBR: During data analyses, the corresponding author Fatema Binte Rasul (FBR) was supported by a sandwich PhD scholarship of USAID's Next Generation of Public Health Experts Program and a DAAD doctoral degree completion grant under the STIBET program. The funders had no role in the design and conduct of the study; collection, analysis, and interpretation of data; writing of the article; or decision to submit the article for publication.

**Competing interests:** The authors have declared that no competing interests exist.

## Introduction

Globally, Non-Communicable Diseases (NCDs) are on the rise. In 2019, NCDs accounted for 1.62 billion Disability Adjusted Life Years (DALYs), amounting to 63.8% of total DALYs [1]. The Global Burden of Disease Study 2015 stated that worldwide, NCDs led to roughly 40 out of the total 56 million deaths that year, equivalent to 70% of total deaths [2]. In 2012, the World Health Organization (WHO) warned that 68% of the world's mortality was attributable to NCDs, and about 75% of total NCD deaths occurred in low- and middle-income countries (LMICs), amounting to 28 million deaths. Worse still, 82% of the total premature deaths attributable to NCDs (deaths before turning 70 years) took place in LMICs [3]. The DALYs for NCDs in LMICs have increased from 34% to 52% over the last decade [4].

NCDs affect health, and economically debilitate countries by affecting the young population's productivity [5]. By 2030, NCDs are predicted to cause a collective loss of $47 trillion [6]. If NCDs are not controlled, LMICs are headed towards an economic loss surpassing 7 trillion US dollars (USD) spanning 2011 to 2025, around 500 billion USD per year [6]. In Southeast Asia, NCDs are the prime killer, accounting for 62% of all deaths, causing around 8.5 million deaths annually [7]. Almost every other NCD death (48% of total NCD deaths) in this region is premature [7], an amount higher than any other WHO regions [8]. As NCDs are causing both a health burden and an economic burden worldwide, they are explicitly addressed in the Sustainable Development Goal (SDG) 3, which aims at decreasing by at least one-third NCD-induced premature mortality via prevention and treatment [9].

In Bangladesh, 67% of total deaths are attributable to NCDs [10]. The Bangladeshi health system is inadequately prepared to handle the challenges of the NCD epidemic, while still struggling with maternal and communicable conditions [11]. A key reason for such inadequacy is the persistence of a policy implementation gap. Latest NCD action plans and strategies include the "Strategic Plan for Surveillance and Prevention of NCDs in Bangladesh" (2011–2015) [11], and the "Multi-Sectoral Action Plan for Prevention and Control of NCDs 2018–2025" [12]. In the current Health, Population and Nutrition Sector Program 2017–2022 (HPNSP), Bangladesh has increased budgets for non-communicable disease control operation plans [12]. However, concrete strategies to translate the ambitions outlined in the aforementioned policies into everyday practice are absent [11]. For example, in Bangladesh, only 30 out of 209 listed essential drugs are for NCDs, many of which are rarely prescribed in practice [13]. In the public sector, only tertiary facilities are equipped to offer NCD services. Alternatively, NCD services are available in the private and in the NGO sector. NCD service provision in Bangladesh is inadequate due to limited health insurance coverage, inadequate resources and insufficient trained personnel in government health facilities [11,13].

While evidence on NCD epidemiology is steadily increasing across LMICs, little is known about people's health-seeking practices [14]. In the Asian context, health-seeking is often a household decision, influenced by factors such as socio-demographics, convenience, service price, position of women in the family, type of illness, perceived quality of care and accessibility [15]. In 2010, a literature review on Asian health-seeking behavior inferred that most studies have been done quantitatively, failing to investigate the actual decision-making process [16]. This review explicitly called for the application of qualitative and mixed methods to explore health-seeking pathways and explain relevant decision-making processes, considering the broader socio-economic and cultural context [16].

In Bangladesh, only a handful of studies have explored factors affecting demand for NCD care [17,18] or adherence to NCD treatment [19]. One study found hypertension to be the most common self-reported condition in central and southern Bangladesh, detecting an increased likelihood of people from higher socio-economic statuses to be diagnosed, and men

to be more likely to seek care from qualified professionals [17]. A second study approached health-seeking behaviors related to NCDs qualitatively, looking at fatal NCDs in women of reproductive age [20]. It found that most women first sought care from semi-qualified providers. Qualified care was sought at a later stage because of the high cost of services, distance, familiarity with semi-qualified providers, and not acknowledging illness severity [20]. A third study adopted a mixed-method approach to explore hypertension and COPD (Chronic Obstructive Pulmonary Disease). Findings indicate that these two conditions impose a financial burden on households and therefore, self-treatment is common in the initial stages of illness. While maintaining a focus on the financial implications of NCDs, this study also revealed a higher propensity to seek formal care among urban residents and suggested a lack of knowledge of care options among rural residents. Providers interviewed explained that people delay seeking care until their NCD deteriorates, largely due to financial barriers and a lack of understanding of the long-term health consequences of NCDs [18].

Our qualitative study aims to fill this gap in knowledge by looking in detail at the health-seeking behavior of people affected by NCDs.

## Methods

### Study aim, design and setting

This qualitative study represents a follow-up to a prior quantitative study that assessed determinants of health-seeking and its related expenditure [21]. Albeit being partially informed by the prior survey [21], this study is exploratory in nature, hence it aims at openly exploring all factors shaping decisions regarding NCD health-seeking behavior in northern Bangladesh, rather than explaining results from the quantitative assessment. Health seeking behavior refers to the series of actions that people undertake with the aim of curing ill-health as per their perception [22]. For this study, we have focused on pathways to care in response to perceived illness [23].

Our study was conducted in the Mithapukur upazila (including one urban and one rural union—Durgapur and Mirzapur) in Rangpur district, northwest Bangladesh. Rangpur is one of the poorest regions in country, with a low literacy rate [24]. We purposely selected the Mithapukur upazila as the study site for our qualitative study, because our prior quantitative assessment [21] had revealed a higher reporting of NCDs in Mithapukur compared to other upazillas, hence facilitating respondent identification and sampling.

Healthcare provision in Rangpur reflects the medically pluralistic context of Bangladesh, and as such, relies on a mixture of public, private and Non-Government Organization (NGO) providers. The public sector has six tiers: ward, union, upazila (subdistrict), district, divisional and national. Primary care facilities are at the ward, union and upazila level, where NCD services are in initial stages of development [25]. At the ward and union level, primary care is provided by outlets known as community clinics and union sub-centers, providing basic screening and referral for NCDs [26]. At sub-district level, NCD corners have been established recently (after this study took place). Medical doctors are available, but can do little in absence of resources for NCD diagnosis and management, and systematic guidelines to offer NCD services [27]. The district level tier provides secondary care through the district hospital. It offers limited diagnostic and curative NCD services [11]. Tertiary care is provided at divisional and national level, through medical college hospitals, post-graduate medical hospitals and specialized health institutes [28]. Overall, there is no streamlined NCD care and no standard preventive and referral protocols in place for NCDs in the public facilities [11]. The 2014 Bangladesh Health Facility Survey reported that only 20% of the country's public health facility could

provide adequate comprehensive NCD services (assessment, prescription, medications and diagnostics), with tertiary facilities being the prime providers [29].

The study area Mithapukur, primarily reliant on subsistence and cash-crop farming, has a population of 527,457 [30]. Nearly half of its population (45%) falls under the national poverty line [31]. The community clinics and union sub-centers of Mithapukur are supposed to refer patients to their upazila health complex (50-bed primary facility) in Durgapur, the urban union. If needed, patients are referred to Rangpur Medical College Hospital (1020-bed tertiary facility) [29,32].

### Data collection strategies and sampling

Between December 2015 and January 2016, we collected data from people affected by NCDs, other members of the community, and healthcare providers using a combination of in-depth interviews (IDIs) and Focus Group Discussions (FGDs). We targeted both community members and healthcare providers to enhance completeness of information and triangulate findings across data sources.

We purposely sampled 25 individuals from the pool who had reported at least one non-communicable chronic condition during the prior quantitative survey [21]. We have included participants with at least one chronic condition, not a healthy individual because our aim was to find out how a person navigates the health system for their health needs when they are affected by a chronic non-communicable disease. This has been a purposive sampling decision, as choosing a group who are already dealing with chronic NCDs increased our chances to elicit more information related to health-seeking for these conditions. To better unravel heterogeneity around health-seeking decisions, we sampled individuals in relation to their previously revealed treatment choice (qualified, semi-qualified, or no treatment/self-care), residency (urban or rural), and sex. Likewise, we purposely selected 21 health providers based on their qualifications (qualified or semi-qualified), affiliation (private or public), and location of practice (urban or rural). Both these individuals and the healthcare providers served as respondents for in-depth interviews. Semi-qualified professionals are any allopathic or traditional providers with some degree of training and experience in primary care, but no specific expertise in CNCDs. For example, medical assistants, village doctors, community health workers, drug-store keepers, and traditional healers. Qualified professionals are health providers who are registered and trained physicians (i.e., MBBS doctors) [33,34]. In addition, we conducted six FGDs, reaching a total of 35 community members. We conveniently sampled FGD participants with support from local grassroots health providers, based on location of residence (urban or rural), sex, and age (middle age and elderly).

The first author (Bangla-speaker, trained medical doctor with public health research experience) and two research assistants (trained Bangla-speaker anthropologists) collected the data. They conducted all interviews and FGDs using pre-tested semi-structured interview guides (S1, S2 and S3 Guidelines), developed beforehand by the research team. The IDI guides for community members affected by NCDs and health providers explored the role of individual factors (such as age and sex), household factors (such as socio-economic status), and contextual factors (such as broader cultural values and social norms) on health-seeking for NCDs and related expenses, perceptions and experiences regarding NCDs, and provider preferences. The IDIs with the people affected by NCDs were mostly conducted at their home, except for one respondent who chose to talk at another place of his choice. The interviews with the health-care providers were conducted mostly at their respective workplaces, a couple were done in their home as per their choice. The FGDs were conducted inside the community, in an open common place where 6–8 people could sit and discuss comfortably. The FGD guides

focused more specifically on contextual factors shaping NCD health responses at community level. All interviews and FGDs were conducted in Bangla, audio-recorded, and transcribed verbatim in Bangla. Only the quotes used in this article were translated into English by the lead author.

**Ethical Approval and consent to participate.**   Ethical approval received- from Ethical Review Committee of BRAC JPG School of Public Health, BRAC University, on December 21, 2015 (Ethics reference no: 71). Interviewers obtained informed written consent from all respondents before the interview was conducted. Furthermore, written permission was obtained from the Upazila Health and Family Planning Officer (UHFPO) of Mithapukur to interview public sector providers from various tiers of health facilities in Mithapukur. The investigators offered a token gift to the NCD-affected respondents and FGD participants in appreciation of their time.

**Consent for publication.**   All participants were asked for consent to use non-attributable quotes in publications during the consenting process. Other than this, no individual data, images or videos of participants have been published.

## Analytic approach

This study's conceptual framework (Fig 1) is rooted in the Andersen conceptual model and considers "Health-seeking behavior for NCDs" as a focal point, which is influenced by individual and household characteristics (including cultural beliefs and values) and contextual factors (including socio-cultural factors, the health system and broader environmental factors) [35].

We applied thematic analysis to distill findings from the transcripts. We proceeded in steps. First, the first and third author (both native Bangla speakers with clinical and public health experience) independently developed codes which reflected the themes explored in the interview guide. Second, they independently coded the transcribed material using the abovementioned codes, but allowed for additional codes to emerge as they proceeded through the analysis. Third, coded material was displayed and the relationships between codes and emerging themes were explored with support from all authors. Discordant interpretations of the

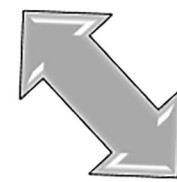
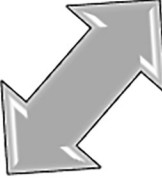

**Fig 1. Conceptual framework: Health-seeking behavior for non-communicable diseases (NCDs).**

findings were resolved via discussion among all authors and/or by returning to the data when necessary. Last, all authors agreed on an emerging interpretation. We conducted the analysis using Nvivo, version 9. To increase the validity of our findings, we applied data (different data sources), methods (different data collection methods), and investigator (different researchers coding) triangulation [36].

## Results

To give voice to community concerns, we purposely report findings in a very descriptive manner and illustrate the major emerging themes (first, individual and household factors, and second, contextual factors and health-seeking experience) using a wealth of direct quotation from respondents' narratives. The respondents' characteristics are given in Tables 1 and 2. Table 3 lists the abbreviations used to describe the respondent attributes.

### Individual and household factors

Respondents reported family, friends, and semi-qualified professionals to be the ones shaping their NCD knowledge and perceptions most. None of the participants received any NCD awareness messages from government. In the context of poor literacy rates, the role of education in health-seeking seemed trivial, as it was not brought up by the respondents. After probing, a few acknowledged that education helped in following prescriptions. Most respondents did not have an understanding that NCDs represent life-long conditions, requiring continuous care. Many found it difficult to acknowledge that NCDs are only controllable, not curable. More than half of all NCD respondents believed that their conditions would be cured if God forgave them, and framed health providers only as a means towards a cure. *"The doctor is a medium, Allah is the one who cures." (Ur, R-5, F, A- 50y).* A handful of respondents expressed

**Table 1. Characteristics of respondents affected by non-communicable disease and focused group discussion (FGD) participants.**

| Characteristics | NCD-affected respondents (N) | FGD participants (N) |
|---|---|---|
| Age group: | | |
| Below 60 years | 20 | 24 |
| Elderly (60 years and above) | 5 | 12 |
| Sex: | | |
| Male | 11 | 18 |
| Female | 14 | 18 |
| Education: | | |
| None | 10 | 28 |
| Primary | 9 | 6 |
| Secondary or above | 6 | 2 |
| Employment status: | | |
| Employed | 6 | 7 |
| Unemployed | 19 | 29 |
| Marital status: | | |
| Married | 24 | 36 |
| Unmarried | 1 | 0 |
| Area of Residence: | | |
| Urban Union-Durgapur | 12 | 24 |
| Rural Union- Mirzapur | 13 | 12 |
| **Total respondents** | 25 | 36 |

**Table 2. Characteristics of healthcare providers.**

| Characteristics | | N |
|---|---|---|
| Qualification | Qualified provider/medical doctors (3) | 3 |
| | Semi-qualified providers (18) | 18 |
| Sex | Male | 14 |
| | Female | 7 |
| Practice Location | Urban | 12 |
| | Rural | 7 |
| | Both urban and rural | 2 |
| Affiliation | Public practice | 3 |
| | Both public and private practice | 5 |
| | Private practice | 9 |
| | NGO-BRAC | 4 |
| | Total respondents | 21 |

that different NCDs require different forms of treatment, as allopathic and other forms of treatment (i.e., homeopathy, traditional) serve different purposes. *"People say doctors cannot treat Baat* [rheumatoid arthritis], *it* [cure] *is in the hands of a traditional healer." (Ur, R-4, F)*

A condition is perceived as existing when symptoms are present, and considered severe when symptoms aggravate to an extent that these symptoms interfere with normal daily activities. At this point, the health-seeking process is initiated. People tend to wait until long after the symptoms appear. No one sought care as soon as the symptoms presented. Many reported discontinuing NCD treatments upon disappearance of their symptoms, as they perceived themselves cured. *"I did not do anything else as I was cured." (Ru, R-2, F, 24y)*. Providers expressed their frustration over this behaviour:

> *"If you think about diabetes. . .they took medicines for two months. . . it was controlled. . . they are not feeling problems. . .they stop taking medicines. . .They came back after 4–5 months. . . with an ulcer, abscess, carbuncle* [complications]. . .*cases of hypertension, asthma, the same! (Ur, QP-2)*

This fact, in addition to people's belief that irrespective of providers, only God can ultimately cure the disease, led respondents to shop from provider to provider in search of a cure. *"It is Allah's will. . . some people get cured. . . such places are there, I am telling you." (Ur, R-8, M, A- 50y)*

**Table 3. Abbreviations for respondent attributes.**

| Characteristics | Abbreviation |
|---|---|
| Urban union | Ur |
| Rural union | Ru |
| Non-communicable disease-affected respondents | R |
| Qualified provider | QP |
| Semi-qualified provider | SQP |
| Focused group discussion | FGD |
| Age | A |
| Male | M |
| Female | F |

Most respondents relied on personal and social contacts to initiate health-seeking. As such, interpersonal relationships with providers and their accessibility appeared pivotal in shaping decisions on health-seeking. This explains why almost all of our sample respondents reported to have first sought care from semi-qualified professionals: they are known in the community, are available round the clock, and provide in-home consultations upon request. Moreover, as members of the community themselves, semi-qualified providers are considered trustworthy. People feel comfortable approaching them, as they feel relatable. The absence of qualified providers in rural areas further reinforces this preference towards semi-qualified providers.

*"I have continued to receive care from him* [village doctor]. *I fell ill for three days. All three days we have called him to our home by mobile. He is our nephew in relation* [indicating acquaintance with him] . . . *His home is also nearby. We have a good relationship with him."* (Ru, R-3, F, A-38y).

*"There are no MBBS doctors here."* (Ru, SQP-Village doctor).

Providers are also aware of the popularity of semi-qualified professionals:

*"An MBBS doctor will not go* [to attend to a patient] *at 12 in the night*! *A village doctor will."* (Ur, SQP-Village doctor).

*"So 90 percent people go to a dispensary first. . . and buy medicines. When they see that this is not benefitting them, then they go to the village doctor. After crossing the barrier to the village doctor, they come to hospital."* (Ur, QP-3)

Interviews also revealed that health-seeking experience differed substantially between men and women. A person's gender appeared to determine whether they enjoy the privilege of autonomous decision-making. All women unanimously mentioned that they needed permission from their "guardian" (male member of the household) to be able to decide on health-seeking, attributing their financial dependency on men. None of our female respondents held a formal job. All female community health workers interviewed confirmed needing permission for their own health-seeking. Several women reported having discontinued treatment due to their husbands' disapproval, and no woman had sought care in spite of an unsupportive family. *"If I were a man, I would not have needed anyone's permission."* (Ur, R-3, A-26y).

Responses from FGDs and interviews with providers confirmed the lack of empowerment expressed by women in individual interviews. *"If a woman wants to override a man's word and do something, she obviously cannot do that. Can she*? *Never*!" (FGD-Ur, F). Additionally, most women reported needing a companion to go health-seeking, and expressed concerns about household chores when leaving to seek care. Inversely, men could decide for themselves, and did not require permission of any sort to seek care: *"I can go* [for treatment] *as soon as I say. All the money is mine, my own."* (Ur, R-11, M, A->60y)

All financially dependent elderly respondents were found to forgo treatment sometimes, as they wished not to trouble their family for their treatment expenses. They believed that they should endure this discomfort as a natural process.

*"How can I get treatment when I have no money?. . . I have one son, who sends some money for medicines, sometimes. When it runs out, I don't want any more."* (Ru, R-9, M, A-70+y)

*"Women are very much neglected! When a mother's age becomes 55 years. . .she cannot be of any help to the household. . .ultimately, she is almost considered as a burden"* (Ur, QP-1)

A few respondents reported fear of facing stigma due to NCDs. Respondents revealed that among their communities, the belief persist that NCDs affect sexual performance, hence reducing one's chances to find a spouse.

*"If it is an unmarried boy or a girl. . .If he or she has asthma, diabetes, or hypertension, these ones are tried and overlooked* [as a diagnosis]. . . *If they* [the spouse to-be's family] *know that he/she has this kind of illness, then maybe. . .there will be problems in the marriage and stuff. . . For this specific reason, these* [conditions]. . .*are usually concealed." (Ur, QP-2)*

*"In the year 1999. . . I had much influence in the society. But nobody would proceed* [for marriage] *when they heard about my diabetes." (Ur, R-9, M, A-> 50y)*

## Contextual factors and health-seeking experience

Geographical accessibility, defined both in terms of actual distance and means of transport, represents a major barrier to NCD care-seeking. The government health workers confirmed that NCD diagnostic tools and medications are only available at the Upazilla Health Complex, and therefore not available at grassroots level (union sub-centers and community clinics).

*"Mirzapur* [rural union] *is very far from here* [Mithapukur's main marketplace], *pretty far! Almost. . . 18–20 kilo*[meters], *maybe. And this. . . Durgapur* [urban union] *that you mentioned? That is close. . . One kilo*[meter] *from where we practice. So, from here, people come fast. . . From 18–20 kilo*[meter] *far away, the road. . .is not pitched. . . The communication system is a bit. . .troublesome" (Ur, QP-3)*

*"As much as we need instruments, we also need medicines. On top of that, there is even a need for specialist doctors!" (Ru, SQP-Union sub-center)*

Respondents indicated the cost of treatment being fundamental in shaping health-seeking behavior, leading one to forgo or delay treatment, discontinue or change providers/the course of treatment. This sentiment was stronger among respondents of poorer households (16 respondents) in our sample. *"I did not go for any treatment for about a year. I did not go due to lack of money." (Ur, R-3, A-26y, F)*. Respondents most often regarded qualified care as an expensive option, motivating their decision not to seek care unless in dire need. When explaining the costs incurred by treatment, narratives show that people incurred substantial expenses while availing semi-qualified care, but did not seem to recognize that.

*"[In] Most cases. . . an economic factor always plays a role. As a result, they* [patients] *take medicine occasionally, taking breaks over and over. . .When they feel well. . .they stop taking their medicines entirely!" (Ur, QP-2)*

*"Now I no longer take that medicine* [prescribed asthma medicine]. . . *This* [self-prescribed steroid injection] *suits my budget better."(Ru, R-4, A-35y, F)*

In contrast, respondents from least poor families (9 respondents) did not single out the importance of cost in availing healthcare, but rather insisted on perceived treatment effectiveness and on provider-patient interactions as aspects determining their satisfaction with provider. People affected by NCDs measure treatment effectiveness with the speed with which their symptoms are alleviated. *"It does not matter if money is spent, I have to stay well." (Ur, R-1, F, A-50y, Diabetes)*

People were found to lean towards patient-centred care. Many expressed that they would like to participate in decision-making concerning their treatment, rather than just receiving a prescription. People seemed to appreciate the opportunity to negotiate or at least express their opinions with a semi-qualified provider, an opportunity that rarely arose in qualified care encounters. Many would discontinue treatment if the provider failed to understand their needs. Both respondents affected by NCDs and providers confirmed that the pluralistic nature of the local healthcare system enables discontinuation and frequent shifts across providers.

*"He* [referring to traditional healer] *is my most favorite. . .He does what I tell him to do. . .That is my main concern. . .I don't have any shortage* [of money]*." (Ur, R-11, M, A->60y)*

*"If they* [qualified care provider] *had done an* X-ray, *then my suspicion would have gone away. . . That person didn't even talk to me. . . I went there to have tests done. But as they did not do tests. . . I didn't even take their medicines*!*" (Ru, R-13, M, A- 50y)*

## Discussion

This study is one of the very first qualitative investigations conducted in Bangladesh aiming at exploring health-seeking behavior for NCDs. As such, this study unravels how individual, household, and systemic elements come together to shape decision-making on health-seeking for NCDs. In particular, our findings revealed how cultural and social elements interact with poor geographical and financial accessibility elements to deter individuals affected by NCDs from seeking care. This evidence is very much needed to identify relevant barriers to access care and design appropriate policies to overcome them.

Our findings reveal that people often do not perceive chronic conditions to be sufficiently severe to motivate care-seeking, unless they significantly affect their daily life. Similarly, our findings also revealed that people often discontinue treatment as soon as the most disturbing symptoms disappear. These findings are aligned with prior evidence from Bangladesh [37], as well as from other countries [38,39], indicating that the severity of illness shapes one's care-seeking decisions. Likewise, findings from India [40] and Kenya [41] also revealed that people seek care only once a condition affects their daily life. The practices captured by our analysis do not appear surprising, considering a context of widespread poverty and lack of financial protection [42]. This forces people to ration limited resources cautiously, given sparce access to reliable sources of information on the health impacts of chronic conditions [11]. Nonetheless, such practices can be especially harmful for NCDs, as chronic conditions usually produce symptoms only at an advanced stage, when the treatment becomes more expensive and the health outcomes worse. Considering that major NCDs (cardiovascular diseases, respiratory disease, cancer, diabetes) are responsible for 80% of all premature deaths [43], measures to alleviate the financial burden should be coupled with information and education campaigns to enlighten people on the health and economic penalties of letting a NCD go untreated.

Moreover, we found that in the absence of access to reliable sources of information on NCDs and treatment options available, religious beliefs interact with false notions on the severity of a condition and direct decisions on care-seeking. More specifically, we noted the common belief that NCDs can be cured only with God's forgiveness, so that health providers only serve as a means to alleviate the presence of symptoms. An earlier Bangladeshi study found similar religious beliefs on sexual health-related health-seeking [44]. Similarly, a study in Kenya reported the belief that people affected by chronic conditions would be healed by prayer [41]. While recognizing the need to respect people's religious conviction and trust in

God, we note the dangerous implications of such beliefs on care-seeking. Our findings highlight the fact that due to a combination of religious beliefs, poor understanding of the etiology of disease and its long-term consequences, and an occasional fear of facing stigma, individuals affected by NCDs tend to delay seeking care. Once they have reached the decision to seek care, they often shop across providers, looking for a satisfying care experience and cure. Such shopping results in discontinuity of care, bearing harmful health and financial consequences in the long run [45–47]. Unlike acute illnesses, it is not possible to fully cure chronic conditions. People require life-long treatment, aiming towards extending their lifespan whilst improving its quality, physically and mentally [48]. For such conditions, shopping across providers poses a challenge to both quality and continuity of care, since it implies that people are receiving uncoordinated and fragmented care from providers of various qualifications. At the same time, this pattern indicates the need for a streamlined NCD care provision plan and points to the need to develop more people-centered healthcare systems [49,50].

In line with the above, our findings also revealed that considerations related to geographical accessibility, cost, and familiarity with a provider interact to determine where a person affected by a chronic condition chooses to seek care. On the one hand, the lack of service provision for NCDs in remote locations and the fear of high costs associated with qualified health-seeking discouraged people from referring first to qualified providers. On the other hand, familiarity with semi-qualified providers, their lower perceived cost, and their accessibility at village level encouraged people to seek them first when in need of care. In this regard, our findings are clearly aligned with prior evidence from Bangladesh, suggesting that semi-qualified providers are often the providers of choice due to their interpersonal skills and closeness to the community [18,20,44]. Interestingly, a recent systematic review has revealed that providers' interpersonal skills are key in shaping patients' satisfaction [51]. Similarly, existing evidence suggests that people skills are crucial in ensuring continuity of care for chronic conditions [41]. The policy implications of our findings are straightforward, calling for a radical restructuring of the health system to ensure availability of patient-centered and low-cost basic NCD screening and treatment at lower levels of care. It will be challenging since Bangladeshi public health outlets have not prioritized primary and secondary preventive care for NCDs at the most basic level [11]. The primary and secondary care facilities need skilled professionals, NCD medications, related equipment and coordinated referrals [11,27]. The latest health bulletin from Bangladesh states that basic supplies for the management of hypertension and diabetes are supposed to be available in public primary healthcare outlets [28]. The government has recently activated NCD corners in selected upazila health complexes. The health providers there acknowledge the role of this initiative in increasing NCD-related literacy, services, and referrals, and say the upazila health complex requires standardized procedures, additional training, diagnostic facilities, record-keeping, and medical supplies [52]. A recent study reported that among primary and secondary health facilities, only 24% and 58% were able to provide basic diagnostic and treatment facilities for cardiovascular diseases and diabetes respectively [53].

The fact that respondents in our sample consistently identified the high cost of care as a major barrier to accessing care reinforces findings from our prior quantitative study [21] and from other studies [18,54], suggesting that OOPE for NCDs in Bangladesh is significant. Such findings are not surprising, since the literature has highlighted in depth how the expected cost of care represents a decisive factor to treatment adherence [55], especially in cases of NCDs, where treatment continues long-term [56,57]. Likewise, the literature has shown that people are more likely to discontinue treatment in the absence of health insurance and sufficient income [55]. A review article on South Asia found that NCD-households are more likely to experience OOPE, catastrophic expenses and impoverishment compared to non-NCD

households [54]. A recent 18-country study that included Bangladesh found that among NCD respondents in low income countries, around 39% of women forewent medicines due to expenses, whereas 13% of men did so [58].

Finally, we would like to note how our findings indicate that cultural values and social norms pertaining to women and their position in society affect their ability to make autonomous decisions about their health-seeking. This lack of empowerment and dependence on male household members has been noted in prior Bangladeshi studies [59–61]. Although the preceding study found no significant gender differences in health-seeking [21], this in-depth exploration unearthed an existing blockade imposed by the patriarchal societal context. This indicates that there might be many more women in need of NCD care than studies have previously shown.

## Study considerations

In spite of its strengths as one of the first rigorously implemented qualitative studies on NCDs in the setting, we need to acknowledge a number of limitations. First, we recognize that data were collected in 2016 and that since then, NCD policies and programs have partially changed. Although NCD corners have been established at subdistrict health facilities, these remain in a very early stage of development [52]. Nonetheless, given the paucity of available evidence [11] and given that the socio-cultural and economic setting has remained largely unchanged, our findings and their implications continue to be relevant.

Second, we recognize that the sample included few respondents with school education, and few elderly respondents, limiting our capacity to make meaningful inferences on the role played by education and health in mediating healthcare seeking, and calling for further research. Similarly, we could not reach any woman with formal employment, hampering us from assessing if and how formal employment can act as an enabling element in NCD-related care decisions.

## Conclusion

This study has looked into the intricacies of health-seeking for NCDs, an aspect not extensively researched in Bangladesh. Our findings highlight delayed and fragmented care seeking for NCDs and the absence of continuity of care, rooting from health and religious beliefs and anticipated expenses. The medically pluralistic context and limited access to trained providers and NCD services negatively impact health-seeking behavior. A popular first choice is semi-qualified providers, due to their easy access and interpersonal skills. Possible solutions emerging from our study are introducing people-centered care, raising NCD awareness and meeting the provider and facility's NCD resource needs. Women and the very poor need special access to NCD care, given a patriarchal context and no social health protection coverage. Further studies could look into the effect of gender, old age, and stigma on NCD health-seeking.

## Supporting information

**S1 Guideline. Semi-structured guideline for in-depth interview with respondent affected by NCD.**
(DOCX)

**S2 Guideline. Semi-structured guideline for in-depth interview with healthcare provider.**
(DOCX)

**S3 Guideline. Guideline for focus group discussion.**
(DOCX)

## Acknowledgments

We acknowledge the support from BRAC JPG School of Public Health and BRAC during field work. We are grateful to the respondents for taking part in this study.

For the publication fee we acknowledge financial support by Deutsche Forschungsgemeinschaft within the funding program "Open Access Publikationskosten" as well as by Heidelberg University.

## Author Contributions

**Conceptualization:** Fatema Binte Rasul, Malabika Sarker, Manuela De Allegri.

**Data curation:** Fatema Binte Rasul.

**Formal analysis:** Fatema Binte Rasul, Malabika Sarker, Farzana Yasmin, Manuela De Allegri.

**Funding acquisition:** Manuela De Allegri.

**Investigation:** Fatema Binte Rasul.

**Methodology:** Fatema Binte Rasul, Malabika Sarker, Manuela De Allegri.

**Project administration:** Fatema Binte Rasul, Malabika Sarker, Manuela De Allegri.

**Resources:** Malabika Sarker, Manuela De Allegri.

**Software:** Manuela De Allegri.

**Supervision:** Malabika Sarker, Manuela De Allegri.

**Validation:** Fatema Binte Rasul, Malabika Sarker, Farzana Yasmin, Manuela De Allegri.

**Visualization:** Fatema Binte Rasul.

**Writing – original draft:** Fatema Binte Rasul.

**Writing – review & editing:** Fatema Binte Rasul, Malabika Sarker, Farzana Yasmin, Manuela De Allegri.

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
