## [Decision Letter · Decision Letter 0]

3 Jan 2022

PGPH-D-21-00988

"The doctor is a medium, Allah is the one who cures": Exploring health-seeking behavior for non-communicable chronic conditions in northern Bangladesh

Dear Dr. Rasul,

Thank you for submitting your manuscript to PLOS Global Public Health. After careful consideration, we feel that it has merit but does not fully meet PLOS Global Public Health’s publication criteria as it currently stands. Therefore, we invite you to submit a revised version of the manuscript that addresses the points raised during the review process.

Dear Authors,

Please address all the comments raised by the reviewers.

We look forward to receiving your revised manuscript.

Kind regards,

Palash Chandra Banik, MPhil

Academic Editor

Journal Requirements:

1. Please provide separate figure files in .tif or .eps format only. Please ensure that all files are under our size limit of 20MB.

For more information about how to convert your figure files please see our guidelines: Once you've converted your files to .tif or .eps, please also make sure that your figures meet our format requirements

2. Your manuscript is missing the following sections: Results. Please ensure these are present, and in the correct order, and that any references to subheadings in your main text are correct. An outline of the required sections can be consulted in our submission guidelines here: https://journals.plos.org/globalpublichealth/s/submission-guidelines#loc-parts-of-a-submission

3. In the online submission form, you indicated that "The qualitative data generated and analyzed during this study are not publicly available due to the data containing information that could compromise research participant’s privacy, but are available on reasonable request. Restrictions may apply.". All PLOS journals now require all data underlying the findings described in their manuscript to be freely available to other researchers, either 1. In a public repository, 2. Within the manuscript itself, or 3. Uploaded as supplementary information.

4. Please amend your detailed Financial Disclosure statement. This is published with the article, therefore should be completed in full sentences and contain the exact wording you wish to be published.

ii). State the initials, alongside each funding source, of each author to receive each grant.

iii). State what role the funders took in the study. If the funders had no role in your study, please state: “The funders had no role in study design, data collection and analysis, decision to publish, or preparation of the manuscript.”

Additional Editor Comments (if provided):

Reviewers' comments:

Reviewer's Responses to Questions

**Comments to the Author**

1. Does this manuscript meet PLOS Global Public Health’s publication criteria? Is the manuscript technically sound, and do the data support the conclusions? The manuscript must describe methodologically and ethically rigorous research with conclusions that are appropriately drawn based on the data presented.

Reviewer #1: Yes

Reviewer #2: No

2. Has the statistical analysis been performed appropriately and rigorously?

Reviewer #1: N/A

Reviewer #2: N/A

3. Have the authors made all data underlying the findings in their manuscript fully available (please refer to the Data Availability Statement at the start of the manuscript PDF file)?

Reviewer #1: Yes

Reviewer #2: Yes

4. Is the manuscript presented in an intelligible fashion and written in standard English?

Reviewer #1: Yes

Reviewer #2: Yes

5. Review Comments to the Author

Reviewer #1: The manuscript describes a synthesis of health seeking approaches in northern Bangladesh population suffering from NCDs. I believe the importance of this paper stems from the applicability of the approach to a large population base, and the challenge we will have in healthcare delivery. This work presents a paradigm that can be broadly and usefully applied.

It is an important topic, needed, and useful summary of the status of service delivery from a certain perspective. The authors, however, need to be bolder and more analytical. This is an opinion piece and a thorough analysis is needed. For example, the financial status of the population surveyed. Including more unemployed and financially poor will result in bias of behavior. A certain view is implied by the authors, but they could be more explicit.

The paper would be both more compelling and useful to a broad readership if the authors moved beyond providing a simple summary of the problems and deeply examined why there is difference in approach in some areas if they included a big sample size with a larger geographical base and then used the evidence they have compiled to suggest a path forward.

I think this oversimplification instigates deeper analysis in other areas as well. I would like to see more examination of the validation requirement beyond the lens of patient health seeking behavior vis a vis availability of resources and healthcare services at the disposal of the segment of population provided by the state in this region.

The authors have tried to capture brilliantly the population beliefs in interview transcripts. The paper relates to in-depth belief of the local populace and thus provide an understanding of the basic problem which can be addressed by the state.

Finally, the takeaway message from this article is well summarized, with suggested directions and immediate challenges to overcome, a call to action and further analysis.

Overall, though this is a timely and needed research. It is well researched and nicely written with rich metaphor. With minor modifications addressing financial status (no details just simplified explanation as below poverty line/ lower middle class, middle class etc. would suffice) this will be a worthwhile review paper. With more significant modification where the authors dig deeper into the complexities and controversies and truly grapple with their implications to suggest a way forward, this could be a very influential paper.

Reviewer #2: Fundamental

1. Why the authors included participants with at least one chronic condition? Why not the healthy individual? The participants with chronic conditions have well orientation with the disease process and management. This orientation suppresses their primary belief and practice. Inclusion of this study group probably overrepresent the study findings/overestimated the all components of analysis.

2. Where the data was collected? At the healthcare facilities or household or any other sites?

3. This reviewer assessed the questionnaire (S2 file, health provider interview guideline) used for healthcare providers. Question number 05 is really irrational or without logic. The question was “At which stage of these diseases do they usually come (early/ advanced stage)?”. Among the 21 healthcare providers, 18 were semi qualified. How this semi-qualified healthcare providers graded a disease into stages as early or advanced? How the authors segregated the healthcare providers into qualified and semi-qualified? This is a vital question in this study as the findings suggested (aligned with global findings) the health-seeking behaviours initiated when symptoms become sever. The findings related to this question will misguide the readers and subsequent policy recommendation.

4. The authors presented “The government health workers confirmed that NCD diagnostic tools and medications are only available at the Upazilla Health Complex…” What percentages of government health workers participated in this study? I have not found any government health workers included in this study group. What the authors mean by ‘Both public and private practice’ as affiliation? Need clarification.

5. Who are the actors participated in this study? Inclusion of actors increases the strength of such study. Did the authors include teachers or religious leaders or any representative of a government health facility?

6. Why the authors avoided to represent their findings based on urban and rural? This is a fundamental issue for a country with fragile health system. Moreover, the health-seeking behaviours largely vary in these two areas. This reviewer like to see the urban-rural variations in terms of ‘health-seeking behaviours.

7. Did the authors assess ‘Knowledge’ about the term ‘Noncommunicable Diseases’? And what are the factors responsible for NCDs? These are the basics that influenced their spiritual action. Based on the questionnaire the authors used, this reviewer assuming ‘knowledge’ in terms of NCD and its risk factors did not asses explicitly. However, the authors tried to find out the sources (family, friends, and semi-qualified professionals) those shaped their knowledge. How the authors interpret their overall findings in this regard? If the participants did not hear the term ‘Noncommunicable Diseases’ in their lifetime, then how their conception will be built up around this entity of disease?

8. What about the limitations of this study? The authors avoided to mention sampling strategy as purposive, small numbers of workshop and separate workshops for both men and women as anticipated that gender-related differences in care seeking were likely. Moreover, there is a gray area about the engagement of local leaders or actors.

Minors

1. Change the title as it is reflecting a specific religious belief. Make it as simple as possible.

2. Change the reference number 1 & replace with recent GBD study

3. Authors have not yet defined the term ‘health-seeking behavior’ in their manuscript? What are the components they considered to assess?

6. PLOS authors have the option to publish the peer review history of their article (what does this mean?). If published, this will include your full peer review and any attached files.

**Do you want your identity to be public for this peer review?** For information about this choice, including consent withdrawal, please see our Privacy Policy.

Reviewer #1: **Yes: **Manish Barman

Reviewer #2: No

---

## [Decision Letter · Decision Letter 1]

27 Apr 2022

Exploring health-seeking behavior for non-communicable chronic conditions in northern Bangladesh

PGPH-D-21-00988R1

Dear Dr. Rasul,

We are pleased to inform you that your manuscript 'Exploring health-seeking behavior for non-communicable chronic conditions in northern Bangladesh' has been provisionally accepted for publication in PLOS Global Public Health.

Best regards,

Palash Chandra Banik, MPhil

Academic Editor

Dear Authors, please address all the concerns if any raised by the reviewers. Thank you again for your good work.

Reviewer Comments (if any, and for reference):

Reviewer's Responses to Questions

**Comments to the Author**

1. If the authors have adequately addressed your comments raised in a previous round of review and you feel that this manuscript is now acceptable for publication, you may indicate that here to bypass the “Comments to the Author” section, enter your conflict of interest statement in the “Confidential to Editor” section, and submit your "Accept" recommendation.

Reviewer #1: All comments have been addressed

Reviewer #3: All comments have been addressed

2. Does this manuscript meet PLOS Global Public Health’s publication criteria? Is the manuscript technically sound, and do the data support the conclusions? The manuscript must describe methodologically and ethically rigorous research with conclusions that are appropriately drawn based on the data presented.

Reviewer #1: Yes

Reviewer #3: Yes

3. Has the statistical analysis been performed appropriately and rigorously?

Reviewer #1: I don't know

Reviewer #3: N/A

4. Have the authors made all data underlying the findings in their manuscript fully available (please refer to the Data Availability Statement at the start of the manuscript PDF file)?

Reviewer #1: Yes

Reviewer #3: Yes

5. Is the manuscript presented in an intelligible fashion and written in standard English?

Reviewer #1: Yes

Reviewer #3: Yes

6. Review Comments to the Author

Reviewer #1: I would like to thank the authors for accepting and incorporating the suggestions. In my opinion the key fundamental concerns have been addressed.

Reviewer #3: First of all, authors have addresses a great topic and demonstrated a vast picture regarding a time burning issue. They have made substantial changes on revision and addressed all the concerns exquisitely . No doubt, the work is in excellent form now. But as like authors, i have a same issue in mind that, already seven years have passed for this data collection and profound steps, policy implications and strategic changes have made for NCD prevention and control. So, this is a real issue and it demands further exploration of health seeking behaviors.

I have noticed irrelevancy in the methods part in few lines. If authors took them into consideration, it will be good enough.

1. I do not understand the necessity of Line 169 to line 177 in the method part. These lines is not that much striking in this method part. Rather, author should put a geographical images of the area for a clear view to the readers.

3. Few operational definitions like - health seeking behavior, semi qualified professionals are somehow putted here haphazardly. Few definitions can be given in a same place with a heading, i think.

2. Whether the authors have used any standards for reporting qualitative research ? I found that part is missing.

Otherwise, Verbatim are properly and excellently placed and discussed all through the write up.

7. PLOS authors have the option to publish the peer review history of their article (what does this mean?). If published, this will include your full peer review and any attached files.

**Do you want your identity to be public for this peer review?** For information about this choice, including consent withdrawal, please see our Privacy Policy.

Reviewer #1: **Yes: **Manish Barman

Reviewer #3: No
